# Disentangled Unsupervised Skill Discovery
# for Efficient Hierarchical Reinforcement Learning

**Jiaheng Hu**
University of Texas at Austin
jiahengh@utexas.edu

**Zizhao Wang**
University of Texas at Austin
zizhao.wang@utexas.edu

**Peter Stone**[†]
University of Texas at Austin, Sony AI
pstone@cs.utexas.edu

**Roberto Martín-Martín**[†]
University of Texas at Austin
robertomm@cs.utexas.edu

## Abstract

A hallmark of intelligent agents is the ability to learn reusable skills purely from unsupervised interaction with the environment. However, existing unsupervised skill discovery methods often learn *entangled* skills where one skill variable simultaneously influences many entities in the environment, making downstream skill chaining extremely challenging. We propose **Disentangled Unsupervised Skill Discovery** (**DUSDi**), a method for learning *disentangled skills* that can be efficiently reused to solve downstream tasks. DUSDi decomposes skills into disentangled components, where each skill component only affects one factor of the state space. Importantly, these skill components can be **concurrently** composed to generate low-level actions, and efficiently chained to tackle downstream tasks through hierarchical Reinforcement Learning. DUSDi defines a novel mutual-information-based objective to enforce disentanglement between the influences of different skill components, and utilizes value factorization to optimize this objective efficiently. Evaluated in a set of challenging environments, DUSDi successfully learns disentangled skills, and significantly outperforms previous skill discovery methods when it comes to applying the learned skills to solve downstream tasks. Code and skills visualization at jiahenghu.github.io/DUSDi-site/.

## 1 Introduction

Reinforcement learning (RL) algorithms have achieved many successes in challenging tasks, including magnetic plasma control [11], automobile racing [54], and robotics [47]. However, applying existing RL algorithms to every new task in a *tabula rasa* manner often results in low sample efficiency that limits RL's broader applicability [18]. Unsupervised skill discovery holds the promise of improving the sample efficiency of Reinforcement Learning, by learning a set of reusable skills through reward-free interaction with the environment that can be later recombined to tackle multiple downstream tasks more efficiently. In practice, prior unsupervised RL skills are represented as a policy that conditions on a skill variable to generate diverse behaviors, and have led to successful and efficient learning of downstream tasks when combined with skill fine-tuning or hierarchical RL skill selection [13, 24, 58].

Despite prior successes, a common limitation of the skills learned by existing unsupervised RL methods is that they are *entangled*: any change in the skill variable causes the agent to induce changes in *multiple dimensions* of the state space simultaneously. Learning to use and recombine these *entangled* skills can be extremely hard for an agent trying to solve downstream tasks, especially in

---

[†]Equal supervision.

38th Conference on Neural Information Processing Systems (NeurIPS 2024).

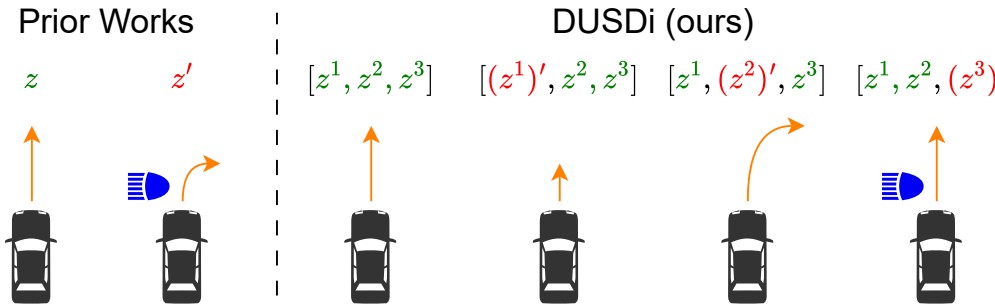

Figure 1: Consider an agent practicing driving skills by learning to control a car's speed (length of orange arrow), steering (curvature of orange arrow), and headlights (blue symbol), **(Left)** previous unsupervised skill discovery methods learn *entangled* skills, where a change in the skill variable can cause all three environment factors to change **(Right)** DUSDi learns *disentangled skills* with concurrent components, where each skill component only affects one factor of the state space, enabling efficient downstream task learning with hierarchical RL.

complex domains like multi-agent systems or household humanoid robots, where the agent needs to concurrently change multiple independent dimensions of the state to complete the task. For example, consider an agent learning to operate a car: if a single skill variable simultaneously changes the speed, steering, and headlights of the car, it will be extremely challenging for the agent to learn how to turn on/off the headlights while keeping the car at the right speed and direction. In contrast, humans naturally have the ability to concurrently and independently adjust the car's acceleration, steering, and headlights based on the car's current speed, surroundings, and lighting conditions. In other words, humans naturally obtain *disentangled* skill components where each component only affects one or few state variables, and can easily recombine these skill components into *compositional skills* [2] to control multiple factors simultaneously.

In this work, we aim to create such a mechanism for artificial agents to learn disentangled skills that facilitate solving downstream tasks. We introduce Disentangled Unsupervised Skill Discovery (DUSDi), a novel method for unsupervised discovery of disentangled skills. A key insight of DUSDi is to take advantage of state factorization that is naturally available in unsupervised RL environments [13, 35, 17] (e.g. speed, direction, and lighting conditions of the car in the driving example; the state of different objects in a household environment). These factored state spaces provide a natural inductive bias we leverage for disentanglement: DUSDi decomposes skills into disentangled components, and encourages each skill component to affect only one state factor while discouraging it from affecting any other factors. To that end, DUSDi designs a novel intrinsic reward for unsupervised skill learning based on mutual information (MI) between disentangled skills and state factors: the learning agent receives high rewards for 1) increasing the MI between a state factor and the skill component assigned to change it, and 2) for decreasing the MI between that skill component and all other state factors.

DUSDi introduces a set of technical innovations to tractably and efficiently optimize the proposed mutual information objective. Once the DUSDi skills are learned, they can be used as the low-level policy in a hierarchical reinforcement learning (HRL) setting to tackle downstream tasks. Compared to using entangled skills, a key benefit of using the disentangled DUSDi skills is that they guarantee more efficient exploration during downstream task learning and therefore often lead to significantly better performance. Furthermore, the structured skill space of DUSDi opens up additional possibilities to inject domain knowledge into the learning process to further improve the efficiency of both skill learning and downstream task learning.

DUSDi is easy to implement and can be integrated into any MI-based unsupervised skill discovery approach. In our experiments, we integrate DUSDi with DIAYN [13] and evaluate the performance on four domains: a 2D agent navigation domain, a DMC walker domain, a large-scale multi-agent particle domain, and a 3D realistic simulated robotics domain. Our experiments indicate that DUSDi can indeed learn disentangled skills, and significantly outperforms other Unsupervised Reinforcement Learning methods on solving complex downstream tasks with HRL.

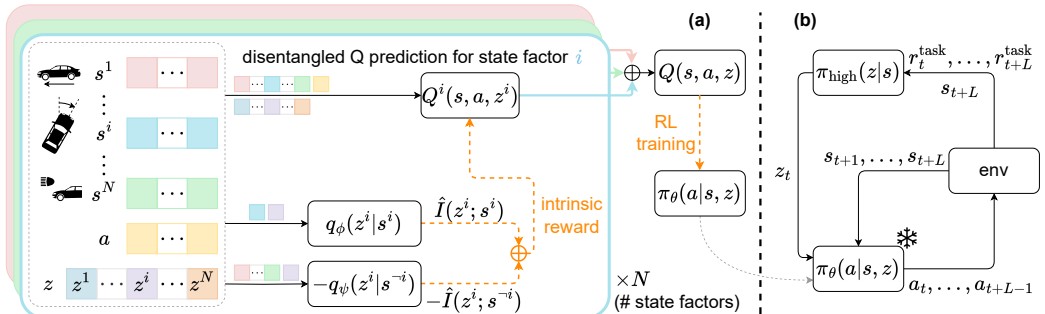

Figure 2: Two learning stages of DUSDi: **(a)** in *disentangled* skill learning stage, DUSDi creates a one-to-one mapping between state factors and skill components — each disentangled skill component $z^i$ only influences state factor $s^i$. DUSDi designs a novel mutual-information-based intrinsic reward to enforce disentanglement and utilize $Q$-value decomposition to learn the skill policy $\pi_\theta$ efficiently. **(b)** in the task learning stage, the skill policy is used as a frozen low-level policy and a high-level policy $\pi_{\text{high}}$ is learned to select skill $z$ for every $L$ steps, by maximizing the task reward $r^{\text{task}}$.

## 2 Preliminaries

**Factored Markov Decision Process (f-MDP)**   In this work, we consider unsupervised skill discovery in a reward-free Factored Markov Decision Process. Following Osband and Van Roy [33], Mohan et al. [32], we define a Factored Markov Decision Process by the tuple $\mathcal{M} = (\mathcal{S}, \mathcal{A}, \mathcal{P})$, where $\mathcal{S} = \mathcal{S}^1 \times \cdots \times \mathcal{S}^N$ is a factored state space with $N$ factors such that each state $s \in \mathcal{S}$ consists of $N$ state factors: $s = (s^1, \ldots, s^N), s^i \in \mathcal{S}^i$. $\mathcal{A}$ is the action space, and $\mathcal{P}$ is an unknown Markovian transition model, $\mathcal{S} \times \mathcal{A} \rightarrow \mathcal{S}$. Notice that a factored state space is often naturally available in domains used by prior works [13, 24, 35, 17, 9, 49] as it can naturally represent environments with separate elements (e.g., objects) that can be changed independently. DUSDi leverages the property that factors often have sparse dynamics dependencies, which opens up the possibility of learning disentangled skills to control the state of each factor. Moreover, many downstream tasks are defined by specific changes in one or a few factors (e.g., changing the state of a single object and not others), which are easier to learn with disentangled skills. In domains with only image-based (unfactored) observations, a factored state space can be extracted using disentangled representation learning or object-centric representation learning methods [31, 20], which we empirically evaluated in Sec. 4.5.

**Mutual-Information-Based Skill Discovery**   Mutual-information-based skill discovery methods, such as the paradigmatic DIAYN [13], specify the skills with a latent variable $z \in \mathcal{Z}$, and learns a skill-conditioned policy $\pi(a|s, z)$. The optimization objective these methods use to learn the skills is to maximize the mutual information (MI) between the state, $s$, and the skill latent variable, $z$: $I(\mathcal{S}; \mathcal{Z})$, which incentivizes the agent to reach diverse and distinguishable states. One popular way to determine the MI, $I(\mathcal{S}; \mathcal{Z})$, is to decompose it as $I(\mathcal{S}; \mathcal{Z}) = H(\mathcal{Z}) - H(\mathcal{Z} \,|\, \mathcal{S})$, where $H$ denotes entropy. Since the skill variable is typically sampled from a fixed distribution, $H(\mathcal{Z})$ can be assumed constant: maximizing $I(\mathcal{S}; \mathcal{Z})$ is thus equivalent to minimizing $H(\mathcal{Z} \,|\, \mathcal{S})$. Following the definition of conditional entropy, $-H(\mathcal{Z} \,|\, \mathcal{S}) = \mathbb{E}_{s,z}[\log p(z|s)]$, DIAYN proposes to approximate $p(z|s)$ with a learned *discriminator* $q(z|s)$ that predicts the skill latent, $z$, given the state, $s$.

After discovering the skills, mutual-information-based methods apply them to learn downstream reward-supervised tasks. Many methods (e.g., DIAYN) adopt a hierarchical RL structure for this second phase, where the skill policy is used as a low-level "frozen" element, and a high-level policy $\pi_{\text{high}}(z|s)$ learns to sequentially activate skill $z$ based on observations. The high-level policy is trained to maximize the provided task reward, $\mathcal{R}$, with $\mathcal{Z}$ as the action space.

## 3 Learning Disentangled Skills with DUSDi

Similar to prior works in unsupervised skill discovery, DUSDi implements a two-stage learning procedure for the agents: in the first phase, DUSDi develops a library of skills without external reward (Sec. 3.1). The key to DUSDi's success is to encourage disentanglement between different skill components through a novel learning objective that restricts the effect of each disentangled skill

**Algorithm 1** DUSDi Skill Learning

---

1: Initialize skill policy $\pi_\theta$, discriminators $q_\phi^i$, $q_\psi^i$ and value function $Q^i$ for each state factor $\mathcal{S}^i$.
2: **for** each skill training episode **do**
3:     Sample skill $z \sim p(z)$.
4:     Collect state transitions with actions from $\pi_\theta(a|s, z)$.
5:     Sample a batch of $(s, a, z)$ from the replay buffer.
6:     **for** $i = 1, \ldots, N$ **do**
7:         Update $q_\phi^i(z^i|s^i)$ and $q_\psi^i(z^i|s^{\neg i})$ with discrimination losses.
8:         Calculate $r^i$ based on Eq. 4
9:         Update $Q^i(s, a, z)$ with reward $r^i$ using SAC.
10:     **end for**
11:     Update $\pi_\theta$ with $Q = \sum_{i=1}^N Q^i$ using SAC.
12: **end for**

---

component to independent factors. In the second phase, DUSDi leverages the learned skills to solve downstream tasks through Hierarchical Reinforcement Learning, achieving higher returns $\sum \mathcal{R}_{\text{task}}$ than methods with entangled skills (Sec. 3.3). In practice, learning disentangled skills in environments with many factors can be challenging. To address this challenge, we introduce improvements to DUSDi's first phase based on Q-function decomposition (Sec. 3.2). We present the entire DUSDi pipeline in Fig. 2, and the pseudo-code in Alg. 1.

## 3.1 Disentangled Skill Spaces and Learning Objective

DUSDi aims to create disentangled skill components that can be easily recombined to solve downstream tasks. To that end, DUSDi proposes a novel factorization of the latent skill conditioning variable, $z$, into $N$ independent disentangled components such that the latent space $\mathcal{Z}$ becomes $\mathcal{Z} = \mathcal{Z}^1 \times \cdots \times \mathcal{Z}^N$. We equate $N$ to the number of state factors and consider $z^i \in \mathcal{Z}^i$ the disentangled skill component that affects state factor $i$. The skill policy $\pi(a|s, z)$ takes in $z \in \mathcal{Z}$, which is a composition of the skill components.

While the factored latent space $\mathcal{Z}$ could be discrete or continuous, we consider discrete skill space in this paper, and discuss how DUSDi can be applied to continuous skills in Appendix A. We can then assume that each disentangled component $z^i$ takes the form of an integer, $z^i \in [1, k]$, resulting in a compositional skill, $z$, with the form of a $N$-dimensional multi-categorical vector with $k^N$ possible values. During skill learning, we independently sample each disentangled component $z^i$ from a fixed uniform distribution $p(z^i)$, similar to previous works [13, 35].

Given this factored skill space, our goal is to learn a skill policy network, $\pi_\theta : \mathcal{S} \times \mathcal{Z} \mapsto \mathcal{A}$, such that each disentangled component $\mathcal{Z}^i$ affects and only affects the value of a state factor, $\mathcal{S}^i$. For each disentangled component and state factor pair $(\mathcal{Z}^i, \mathcal{S}^i)$, we encourage diverse and distinguishable behaviors by maximizing their mutual information $I(\mathcal{S}^i; \mathcal{Z}^i)$. While this objective enables a disentanglement skill component to affect the corresponding factor, it does not restrict the component from affecting other factors. This is undesirable since the resulting skill components would still be entangled in their effects. To prevent that, we propose to ensure that each skill component, $\mathcal{Z}^i$, minimally affects the rest of the state factors, $\mathcal{S}^{\neg i}$, where $\mathcal{S}^{\neg i}$ denotes the subspace formed by all other state factor spaces except $\mathcal{S}^i$: $\mathcal{S}^1 \times \ldots, \mathcal{S}^{i-1} \times \mathcal{S}^{i+1} \times \cdots \times \mathcal{S}^N$. Specifically, we incorporate an entanglement penalty to minimize, $I(\mathcal{S}^{\neg i}; \mathcal{Z}^i)$, which corresponds to the mutual information between a skill component and all other state factors that it should not affect.

Formally, the skill policy aims to maximize the following objective:

$$\mathcal{J}(\theta) = \sum_{i=1}^N I(\mathcal{S}^i; \mathcal{Z}^i) - \lambda I(\mathcal{S}^{\neg i}; \mathcal{Z}^i), \tag{1}$$

where $\lambda < 1$ is a hyperparameter that controls the importance of the entanglement penalty relative to the skill-factor association. We restrict $\lambda$ to be smaller than one for the following reason: in some environments, due to intrinsic dynamical dependencies between state factors themselves, controlling

a state factor, $\mathcal{S}^i$, has to introduce some association between $\mathcal{Z}^i$ and other factors in $\mathcal{S}^{\neg i}$, e.g., when controlling an object whose manipulation requires the agent to use other objects as tools. In these cases, as the policy learns to maximize the MI between a skill and a factor, $I(\mathcal{S}^i, \mathcal{Z}^i)$, the MI with other factors, $I(\mathcal{S}^{\neg i}; \mathcal{Z}^i)$, may also increase. For these cases, the use of $\lambda < 1$ will ensure that the entanglement penalty does not overpower the association reward, and the policy is still incentivized to learn disentangled skill components that change $S^i$ distinguishably while introducing minimal changes on other factors. In practice, we simply set $\lambda = 0.1$ in all our experiments.

**Optimizing DUSDi's Objective:** Directly maximizing the objective in Eq. 1 is intractable. Alternatively, we propose to approximate the objective using a variational lower bound of the mutual information [1]:

$$I(\mathcal{S}^i; \mathcal{Z}^i) = H(\mathcal{Z}^i) - H(\mathcal{Z}^i|\mathcal{S}^i) \geq C + \mathbb{E}_{z,s} \log q_\phi^i(z^i|s^i), \tag{2}$$

where $C$ represents the constant value of $H(\mathcal{Z}^i)$, the entropy of the prior distribution over the skill latent variable, which does not change during training, and $q_\phi^i$ is a variational distribution.

Similarly, we can approximate the MI in the entanglement penalty by:

$$I(\mathcal{S}^{\neg i}; \mathcal{Z}^i) \geq C + \mathbb{E}_{z,s} \log q_\psi^i(z^i|s^{\neg i}), \tag{3}$$

where $q_\psi^i$ is another variational distribution[3]. Importantly, when these $q$ approximations perfectly recover the posterior distribution of $z^i$, we obtain equality in Eq. 2 and Eq. 3. In DUSDi, we implement the variational distributions, $q_\phi$ and $q_\psi$, as neural network discriminators mapping input state factor(s) to the predicted disentangled component values, $z^i$.

To optimize $\mathcal{J}(\theta)$, we alternate between two steps: 1) performing variational inference to train the discriminators $q_\phi^i$ and $q_\psi^i$ through gradient ascent, and 2) using $q_\phi^i$ and $q_\psi^i$ to learn a disentangled skill policy $\pi_\theta$ through RL by maximizing the following intrinsic reward approximating Eq. 1:

$$r_z(s, a) \triangleq \sum_{i=1}^{N} q_\phi^i(z^i|s^i) - \lambda q_\psi^i(z^i|s^{\neg i}) \tag{4}$$

Interestingly, the decomposed nature of our intrinsic reward allows a convenient avenue for shaping skill behaviors based on domain knowledge. In particular, we can restrict a state factor $s^i$ to only take certain values by constraining $q_\phi^i(z^i|s^i)$ accordingly. While not the main focus of this work, we briefly explore this further optimization enabled by DUSDi in Appendix H.

## 3.2 Accelerating Skill Learning through Q Decomposition

When using reinforcement learning (RL) to optimize the intrinsic reward function defined in Eq. 4, standard RL algorithms treat the reward function as a black box and learn a single value function from the mixture of intrinsic reward terms. While this approach may be sufficient for environments with few state factors, doing so for complex environments with many state factors (large $N$) often leads to suboptimal solutions. A key reason is that the mixture of $2N$ reward terms leads inevitably to high variance in the reward, making the value of the Q function oscillate. Furthermore, the sum of reward terms obscures information about each term's value, which hinders credit assignment.

DUSDi overcomes this issue by leveraging the fact that the intrinsic reward function in Eq. 4 is a linear sum over terms associated with each disentangled component. Thanks to the linearity of

---

[3]While lower-bounding $\mathcal{J}(\theta)$ requires upper-bounding $I(\mathcal{S}^{\neg i}; \mathcal{Z}^i)$, we stick with the variational lower bound for this term because of the complexity in upper bounding MIs [41].

expectation, we can decompose the Q function into $N$ disentangled Q functions as follows:

$$Q_\pi(s, a, z) = \mathbb{E}_\theta[\sum_{t=0}^{\infty} \gamma^t r_t]$$

$$= \mathbb{E}_\theta[\sum_{t=0}^{\infty} \gamma^t \sum_{i=1}^{N} q_\phi^i(z^i|s^i) - \lambda q_\psi^i(z^i|s^{\neg i})]$$

$$= \sum_{i=1}^{N} \mathbb{E}_\theta[\sum_{t=0}^{\infty} \gamma^t (q_\phi^i(z^i|s^i) - \lambda q_\psi^i(z^i|s^{\neg i}))]$$

$$= \sum_{i=1}^{N} Q^i(s, a, z) \tag{5}$$

where $Q^i$ represents each disentangled Q function, one for each disentangled component. The disentangled Q functions can be then updated only with their corresponding intrinsic reward terms, $r^i \triangleq q_\phi^i(z^i|s^i) - \lambda q_\psi^i(z^i|s^{\neg i})$. During policy learning, we sum all disentangled Q functions together to recover the global critic, $Q_\pi$, as shown in Fig. 2 (a), top. Compared to learning $Q_\pi$ directly from all $2N$ reward terms, learning disentangled Q functions significantly reduces reward variance, allowing $Q_\pi$ to converge faster and more stably.

### 3.3 Downstream Task Learning

Similar to Eysenbach et al. [13], in DUSDi we utilize hierarchical RL to solve reward-supervised downstream tasks with the discovered skills, as depicted in Fig.2 (b). The skill policy, $\pi_\theta : \mathcal{S} \times \mathcal{Z} \to \mathcal{A}$, acts as the low-level policy and is kept constant while a high-level policy, $\pi_{\text{high}} : \mathcal{S} \to \mathcal{Z}$, learns to select which skill to execute for $L$ steps using the skill latent variable, $z$. Thus, the skill latent conditioning space, $\mathcal{Z}$, acts as the action space of the high-level policy, $\pi_{\text{high}}$. As extensively evaluated in our experiments, without any additional "ingredient", performing downstream task learning in the action space formed by DUSDi skills often results in significantly superior performance compared to an action space formed by entangled skills. We show that the superior performance of DUSDi can be explained by **more efficient exploration** when using the DUSDi skills for hierarchical RL, which we elaborate on in Appendix B, through analyzing the benefits and search complexity of DUSDi's skill space over DIAYN's.

Depending on the nature of the downstream tasks, we can often take further advantage of the disentangled skills learned by DUSDi through leveraging its structure. One such scenario is when the downstream task has a composite reward function consisting of multiple terms. Previous works [16, 46] have shown that when the causal dependencies from action dimensions to reward terms are available (e.g., the reward for speed only depends on actions that affect speed), one can use Causal Policy Gradient (CPG) to decompose the policy update (e.g., only the "speed actions" get updated by the speed reward) and greatly improve sample efficiency, especially when the dependencies are sparse. In downstream task learning, with an action space (of the high-level policy) consisting of the skills learned by DUSDi, we have a convenient way of applying causal policy gradient, where the causal dependencies between the action dimensions (i.e., skill components) and reward terms are often sparse and can be easily obtained by examining the state factor that a skill component is associated with, which we evaluate empirically in Sec. 4.6.

## 4 Experimental Evaluation

In the evaluation of DUSDi, we aim to answer the following questions: **Q1**: Are skills learned by DUSDi truly disentangled (Sec. 4.2)? **Q2**: Can Q-decomposition improve skill learning efficiency (Sec. 4.3)? **Q3**: Do our disentangled skills perform better when solving downstream tasks compared to other unsupervised reinforcement learning methods (Sec. 4.4)? **Q4**: Can DUSDi be extended to image observation environments (Sec.4.5)? **Q5**: Can we leverage the structured skill space of DUSDi to further improve downstream task learning efficiency (Sec.4.6)?

Table 1: Evaluation of skill disentanglement based on the DCI metric, shown as mean and standard deviation across skill policies trained with 3 random seeds.

| | 2D Gunner | | Multi-Particle | | iGibson | |
|---|---|---|---|---|---|---|
| | DUSDi (ours) | DIAYN-MC | DUSDi (ours) | DIAYN-MC | DUSDi (ours) | DIAYN-MC |
| Disentanglement ($\uparrow$) | $\mathbf{0.864} \pm 0.018$ | $0.016 \pm 0.002$ | $\mathbf{0.705} \pm 0.037$ | $0.002 \pm 0.000$ | $\mathbf{0.833} \pm 0.022$ | $0.017 \pm 0.006$ |
| Completeness ($\uparrow$) | $\mathbf{0.864} \pm 0.017$ | $0.024 \pm 0.004$ | $\mathbf{0.750} \pm 0.041$ | $0.003 \pm 0.000$ | $\mathbf{0.834} \pm 0.021$ | $0.019 \pm 0.005$ |
| Informativeness ($\uparrow$) | $\mathbf{0.897} \pm 0.012$ | $0.821 \pm 0.010$ | $\mathbf{0.849} \pm 0.052$ | $0.791 \pm 0.032$ | $\mathbf{0.854} \pm 0.006$ | $0.752 \pm 0.015$ |

## 4.1 Evaluation Environments

Previous works [13, 34, 35, 44, 23] extensively rely on standard RL environments such as DMC [48] and OpenAI Fetch [4] to evaluate unsupervised RL methods. However, unlike previous unsupervised skill discovery methods, DUSDi focuses on learning a set of disentangled skill components that can be concurrently executed and re-combined to complete downstream tasks. As such, it only makes sense to examine the performance of DUSDi in challenging tasks that require concurrent control of many environment entities (e.g. multi-agent systems, complex household robots). Previous environments lack this property: in DMC for example, while the state and action space can be very complex, the predominant downstream tasks are just to move the center-of-mass of the agent to different places. In such cases, there is no need for concurrent skill components, and therefore we do not expect large gains from using DUSDi's disentangled skills. Nevertheless, we include an evaluation on the **DMC-Walker** [48] environment to demonstrate that our method is also applicable to those environments, but focus the majority of our evaluation on environments that DUSDi is designed for, including **2D Gunner**, **Multi-Particle** [30], and **iGibson** [26].

The 2D gunner is a relatively simple domain, where a point agent can navigate inside a continuous 2D plane, collecting ammo and shooting at targets. Multi-Particle is a multi-agent domain modified based on [30]. In this domain, a centralized controller simultaneously controls 10 heterogenous point-mass agents to interact with 10 stations, where each agent can only interact with a specific station. We evaluate in this domain to test the scalability of our methods to a large number of state factors. iGibson [26] is a challenging simulated robotics domain with the same action space and complexity as real-world robots, where a mobile manipulator can navigate in a room, inspect the room using its head camera, and interact with electric appliances in the room by pointing a remote control to them and switching them on/off. We evaluate in this domain to examine whether our method can handle home-like environments with complex dynamics. We provide visualizations and additional information about each of the environments in Appendix C.

## 4.2 Evaluating Skill Disentanglement

First, we examine whether the skills learned by DUSDi are truly disentangled (**Q1**) using the DCI metric proposed by Eastwood and Williams [12]. The DCI metric consists of three terms, namely **disentanglement**, **completeness**, and **informativeness**, explained in detail in Appendix F. In the original work, measuring DCI requires knowing the ground truth generative factors. In our case, the generative factors are simply the state factors, and we only need to discretize the value of each state factor to make it compatible for evaluation. For each method on each domain, we collect $100K$ rollout steps using the learned skill policy, $\pi(s, z)$, where the skill is (re)sampled from the uniform prior distribution, $p(z)$, every 50 steps. These (state, skill) pairs are then used to calculate DCI.

We compare against **DIAYN-MC** (Multi-channel DIAYN) that uses the same skill representation as DUSDi but optimizes the DIAYN objective of $I(\mathcal{S}; \mathcal{Z})$, and show results in Table 1. Unsurprisingly, DUSDi significantly outperforms DIAYN-MC, especially on Disentanglement and Completeness, across all three environments. These results indicate that DUSDi learns truly disentangled skills, enabling efficient downstream task learning, as we will show in Sec. 4.4. We encourage the readers to visit our project website for a qualitative visualization of the learned skills.

## 4.3 Evaluating Skill Learning Efficiency with Q-decomposition

To examine the importance of Q-decomposition (**Q2**), we measure the performance of optimizing the DUSDi objective during skill learning with and without a decomposed Q network. We compare the classification accuracy of the skill discriminators $q_\phi^i(z^i|s^i)$, averaged over all skill channels, which

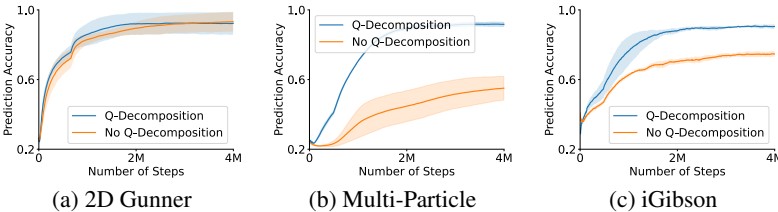

|  (a) 2D Gunner | (b) Multi-Particle | (c) iGibson |

Figure 3: Evaluation of the effect of Q-decomposition in skill learning. The plots depict the mean and standard deviation of accuracy ($\uparrow$) when predicting the skill component $z^i$ based on the state factor $s^i$, computed across 3 training processes. The higher prediction accuracy indicates that the policy learns to control more state factors in more distinguishable ways, leading to more efficient downstream task learning.

indicates progress towards discovering diverse and distinguishable skills, with higher accuracy being better. We depict our results in Fig. 3. We observe that Q-decomposition has a similar performance to the regular Q network in the simplest 2D gunner domain, but significantly outperforms the regular Q network in domains with more state factors (Multi-Particle) and more complex dynamics (iGibson), suggesting that Q-decomposition is necessary for scaling towards complex domains.

## 4.4 Evaluating Downstream Task Learning

The promise of DUSDi is to incorporate disentanglement into skills so that the skills can be effectively used in downstream task learning. Therefore, the most critical evaluation of our work focuses on comparing the performance of different unsupervised RL methods on task learning (**Q3**). We compare against existing state-of-the-art unsupervised reinforcment learning algorithms, including **DIAYN** [13], **CIC** [24], **CSD** [35], **METRA** [36], **ICM** [37], **RND** [5], **ELDEN** [51], and **Vanilla RL** [14], where these baselines are further explained in Appendix E.

Similar to the evaluation setting in the URLB benchmark [23], we allow each method to train for 4 million steps without access to reward (i.e., pretraining phase) before the reward is revealed to the agent and the downstream learning takes place. During the pre-training phase, all methods use soft actor-critic (SAC) [14] to optimize the intrinsic reward. For all skill discovery methods (i.e., DUSDi, DIAYN, CIC, CSD, METRA), a skill-conditioned policy, $\pi_\theta(a|s, z)$, is learned during the pretraining phase. During downstream learning, the skill network is fixed, whereas an upper policy, $\pi_{\text{high}}(z|s)$, is trained using proximal policy optimization (PPO) [43] to optimize the task reward. Similar to previous works [13, 44], we omit proprioceptive states from the MI optimization for all skill discovery methods to facilitate more meaningful explorations. For exploration methods (i.e., RND, ICM, ELDEN), a policy $\pi_\theta(a|s)$ is learned during the pretraining phase on intrinsic reward and fine-tuned using the task reward during the downstream learning phase. The hyperparameters are specified in Appendix G.

We evaluate all methods in four environments and 13 downstream tasks, detailed in Appendix D. The results are depicted in Fig. 4. As expected, DUSDi performs similarly to previous unsupervised RL methods in the DMC walker environment due to the simplicity in terms of its downstream objectives (all related to center-of-mass locomotion), but significantly outperforms all previous methods on domains where downstream tasks require coordinative control of multiple state factors. The most crucial comparison is between DUSDi and DIAYN. DIAYN is a special case of DUSDi where there is only one state factor (consisting of the entire state) and one skill component. Therefore comparing against DIAYN offers a straightforward examination of the effect of disentangled skills for downstream task learning. DUSDi significantly outperforms DIAYN in all downstream tasks, demonstrating the effectiveness of using disentangled skills. In general, we found exploration-based methods to be less capable than skill discovery methods, possibly due to their lack of temporal abstraction. CIC performs very poorly, likely because the CIC objective does not explicitly encourage distinguishable skills and instead generates the intrinsic reward solely based on state entropy, making it very hard for the upper policy to select the right skill. This result again shows the importance of having a proper skill representation. DUSDi also outperforms CSD and METRA on most downstream tasks, especially on the more complex and high-dimensional domains, like Multi-Particle. This superiority is perhaps surprising considering that in our experiments, DUSDi only relies on the simple DIAYN-style intrinsic reward for skill discovery, but further demonstrates the importance of learning a disentangled skill space. It is important to notice that many techniques proposed to improve skill discovery quality

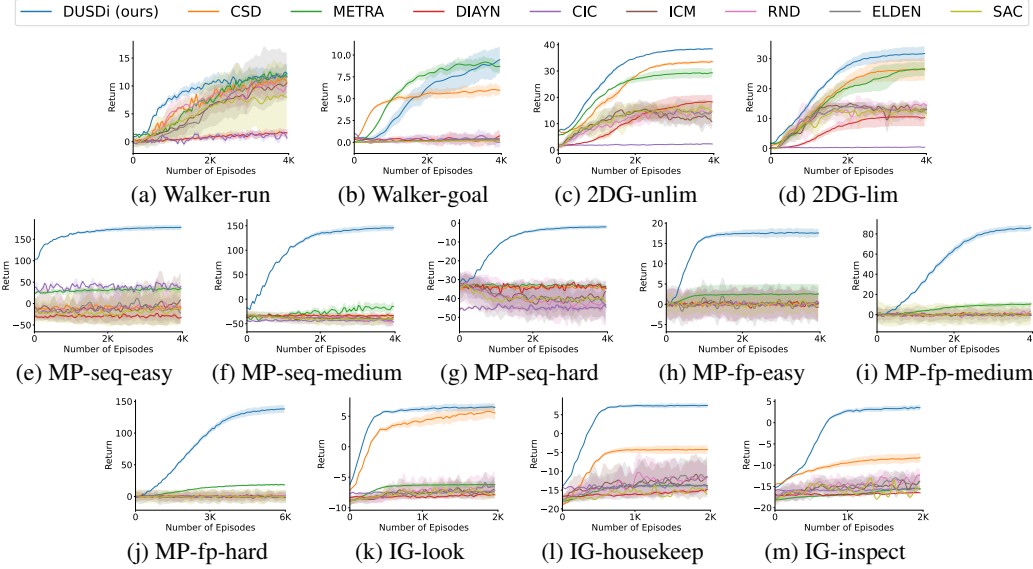

Figure 4: Training curves of DUSDi and baselines on multiple downstream tasks (reward supervised second phase). The plots depict the mean and standard deviation of the return of each method over 3 random seeds. DUSDi outperforms all baselines that learn entangled skills, converging faster and to higher returns.

(e.g., Baumli et al. [3], Zhao et al. [57]), can be seamlessly incorporated into DUSDi. Therefore, we expect our method to perform even better as new advances are made in unsupervised skill discovery.

## 4.5  Extending DUSDi to Image Space

Although this paper primarily focuses on applying DUSDi to factored state space, we can straightforwardly extend it to image space through existing works in factored / object-centric representation learning [29, 20, 53, 28, 55] (**Q4**). We empirically illustrate this capability in the Multi-Particle environment, where we replace the low-dimensional state observation with $64 \times 64$ image observations. Specifically, we first pretrain an object-centric encoder following Yang et al. [55], and then use our method on top of the extracted representation to learn disentangled skills. Hence, essentially, the skill policy uses images as observation. As shown in Fig. 5, when learning from image observation, DUSDi achieves similar performance to learning from state space, whereas the baseline methods are unable to learn these two tasks even when learning from the low-dimensional state space as in Fig. 4.

## 4.6  Leveraging Structure of DUSDi Skills

While DUSDi can already learn downstream tasks quite efficiently, it is possible to further improve the sample efficiency of downstream task learning through leveraging the structured skill space of DUSDi (**Q5**), as described in the second paragraph of Sec.3.3. Specifically, we apply Causal Policy Gradient [16] to the Multi-Particle domain, where the causal dependencies between state factors and reward terms are easy to identify. We present our results in Fig. 6, where the sample efficiency of downstream task learning is greatly improved thanks to the structured skill space of DUSDi.

## 5  Related Work

**Unsupervised Skill Discovery**  In unsupervised skill discovery, the goal of an agent is to learn task-agnostic skills without external rewards. To learn such skills, previous methods propose various forms of intrinsic reward: (1) maximizing the mutual information between visited states and the skill variables [13, 44, 6, 24], (2) maximizing the traveled distance along the direction specified by the skill variables [34–36], (3) learning to reach a diverse set of goals [52, 40, 38]. These skills can be used to boost the sample efficiency of downstream task learning, for example, (1) using hierarchical RL where a high-level policy learns to select which skill to execute [13], or (2) using the skill policy to initialize the task solving policy and then fine-tuning it [24].

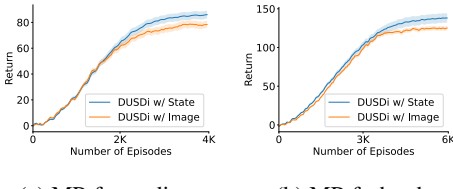

(a) MP-fp-medium      (b) MP-fp-hard

Figure 5: Performance of DUSDi with image observations on two multi-particle downstream tasks over three random seeds. With the help of disentangled representation learning, DUSDi effectively learns skills based only on image observations and leverages the skills to solve challenging downstream tasks where baseline methods fail.

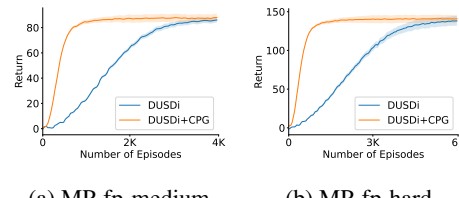

(a) MP-fp-medium      (b) MP-fp-hard

Figure 6: Performance of DUSDi in two multi-particle downstream tasks when combined with Causal Policy Gradient (CPG, orange). The disentangled skills of DUSDi provide opportunities for leverage structure and speed up downstream task learning, greatly improving the sample efficiency when learning downstream tasks.

**State Space Factorization in RL**  In RL, there is a long history of leveraging state factorization, including learning a world model between state factors for planning [22, 50], augmenting data [39], and providing intrinsic rewards [42, 17, 9]. Relevant to our work are skill discovery methods that learn to either reach a goal for each controllable object [19, 9] or achieve interactions between a pair of specified objects [8]. Though these methods achieve disentanglement by influencing one or a pair of objects during a skill, they do not apply to tasks that require controlling multiple objects simultaneously, like driving where we need to control the car's speed and heading directions at the same time. In contrast, our method can combine disentangled skill components into concurrent skills [10] to solve a wide range of tasks.

**Disentanglement in Skill Learning**  Inspired by the benefits of compositionality, disentanglement has been extensively studied, mainly in learning image representations [28]. There are a few works investigating disentanglement in unsupervised skill discovery. Lee et al. [25] consider a special case of disentangled skills — for a multi-arm robot, learning independent skills for each arm. However, they rely on manually factored action spaces which is an assumption that often limits the behavior of the agent. Kim et al. [21] encourage the disentanglement between different dimensions of the skill variable by regularizing it with $\beta$-VAE objective [15], but Locatello et al. [27] point out that such regularization is impossible to achieve disentanglement. To learn disentangled skills, Song et al. [45] learns a decoder from skill variables to state trajectories and their generation factors, which is then used to train the skill policy through imitation learning. However, their training of the decoder requires pre-collected trajectories and corresponding generation factors, whereas our method is fully unsupervised with no expert data.

## 6    Conclusion

We present DUSDi, an unsupervised skill discovery method for learning disentangled skills by leveraging the factorization of the state space. DUSDi designs a skill space that exploits the factorization of the state space and learns a skill-conditioned policy where each sub-skill affects only one state factor. DUSDi enforces disentanglement through an intrinsic reward based on mutual information, and shows superior performance on a set of downstream tasks with naturally factored state spaces compared to baselines and state-of-the-art unsupervised RL methods.

One limitation of DUSDi is the assumption of access to a factored state space. While a factored state space is naturally available in many existing RL environments, and can be extracted from images as we have shown in our experiment (Sec. 4.5), we believe that future advances in disentangled representation learning will greatly broaden the applicability of DUSDi towards partially observable, pixel-based environments. Secondly, DUSDi primarily focuses on learning a structured skill space for more efficient downstream learning, and its exploration capability during skill learning is largely determined by the specific algorithm used to optimize for our mutual information objective. While we used DIAYN [13] in this work due to its simplicity, it would be interesting to examine extending the idea of learning disentangled skills to other skill discovery methods, e.g., Zhao et al. [57], Laskin et al. [24], including those that are not based on mutual information [35, 56].

**Acknowledgements** This work took place at the Learning Agents Research Group (LARG) and the Robot Interactive Intelligence Lab (RobIn) at UT Austin. RobIn is supported in part by DARPA TIAMAT program (HR0011-24-9-0428). LARG research is supported in part by NSF (FAIN-2019844, NRT-2125858), ONR (N00014-18-2243), ARO (W911NF-23-2-0004), Lockheed Martin, and UT Austin's Good Systems grand challenge. Peter Stone serves as the Executive Director of Sony AI America and receives financial compensation for this work. The terms of this arrangement have been reviewed and approved by the University of Texas at Austin in accordance with its policy on objectivity in research.

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

## A  Extension to Continuous Skills

For continuous skills, we can simply define each skill component $z^i$ as a m-dimensional continuous vector (and therefore the length of the skill would be $m \times N$, where N is the total number of skill components). Now we can define the prior distribution for each skill component distribution $p(z^i)$ as an m-dimensional continuous uniform distribution (e.g. $\mathcal{U}[-1, 1]$). Notice that our objective remains unchanged, as the MI is well-defined no matter whether the skill is discrete or continuous. Lastly, we need to change the output head of our skill prediction networks $q$, such that instead of outputting a categorical distribution, it will output a continuous probability distribution (e.g. a multivariate Gaussian distribution with diagonal covariance).

## B  Entangled vs. Disentangled Skill Components for Policy Learning

Compared to entangled skills, the advantages of using disentangled components mainly reside in an easier exploration in the skill space. For skill spaces of equivalent capacity, the DIAYN latent skill variable is a *single* integer between 1 and $k^N$, and the DUSDi skill variable is a $N$-dimensional vector with each dimension representing a disentangled component with $k$ possible values. In this section, we analyze the benefits and search complexity of DUSDi's space over DIAYN's for two main cases: when there are no dynamical dependencies between state factors (optimal case for disentangled components) and where there are intrinsic dependencies between state factors.

**State Factors without Dynamical Dependencies:**   In this case, for DIAYN to find the correct skill to execute at the current time step, in the worst case, it needs to iterate through all skills, resulting in 1-step exploration sample-efficiency of $O(k^N)$. In contrast, for DUSDi, as disentangled components are independent of each other, with one skill trial, the agent can simultaneously observe the effects of setting each disentangled component as $\mathcal{Z}^i = z^i$. Hence, for an intelligent agent, to understand the effects of each disentangled component at the current state, it only needs to sweep through each disentangled component space with $k$ trials (e.g., setting all disentangled components $\mathcal{Z}^i = 1, \dots, k$). After that, as the effects of each disentangled component are independent, by compositing disentangled components in novel ways, the agent has the ability to imagine the effects of all skills, leading to $O(k)$ exploration efficiency.

**State Factors with Dynamical Dependencies:**   When there are dynamical dependencies, we denote $PA^i$ as *parent* indices of state factors that $\mathcal{S}^i$ depends on, e.g., when moving a mouse ($\mathcal{S}^i$), $\mathcal{S}^{PA^i}$ denotes the hand. In such cases, the effect of $\mathcal{Z}^i$ is conditioned on the value of $\mathcal{Z}^{PA^i}$, and we need to iterate through all $(\mathcal{Z}^i, \mathcal{Z}^{PA^i})$ pairs to observe all possible influences on $S^i$. As a result, the exploration is constrained by the state factor with the largest number of parents. Denoting $|PA^i|$ as the number of parent factors for $\mathcal{S}^i$, the exploration sample-efficiency is $O(k^{1+\max_i |PA^i|})$. We can see that the $O(k)$ efficiency when there is no dynamical dependencies is a special case of $\max_i |PA^i| = 0$. Despite lower efficiency than $O(k)$, in many environments, the dynamics of each state factor only depend on a small number of other factors, i.e., $\max_i |PA^i| \ll N$. Hence, exploration with disentangled components is still more sample-efficient than using entangled skills.

## C  Environment Details

We test DUSDi on four environments, where a visualization of each of the environments is presented in Fig. 7.

**2D Gunner:** Shown in Fig. 7 (a), the blue star marks the position of the agent, the blue line marks its shooting direction, the red diamond marks ammo location, and the orange cross marks the target position. The agent has a 7-dimensional observation space, consisting of 3 state factors: [Agent Position, Ammo State, Target State]. The action is 5-dimensional, 2 for agent movement, 2 for ammo pickup, and 1 for shooting direction.

**DMC-Walker:** Shown in Fig. 7 (b), a 6 degree-of-freedom robot can locomote on a 2D plane through joint motions. The agent has a 26-dimensional observation space consisting of 3 state factors: [Body Position, Body Velocity, Robot Proprioception].

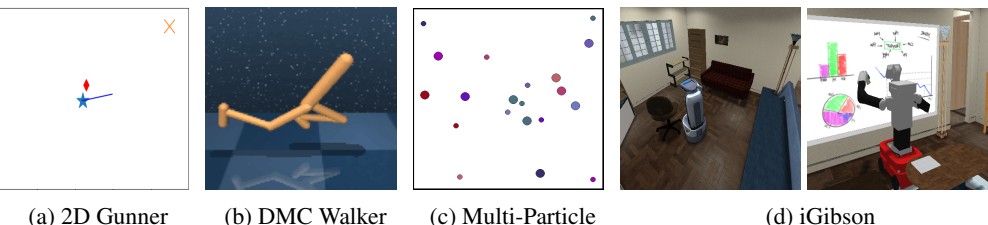

| (a) 2D Gunner | (b) DMC Walker | (c) Multi-Particle | (d) iGibson |

Figure 7: Environments Visualization

**Multi-Particle:** Shown in Fig. 7 (c), the agents are marked by small circles, while the stations are marked by large circles. Only stations and agents of the same color can interact with each other. The Multi-Particle environment has a 70-dimensional observation space, consisting of 20 state factors. The state factors include states for each landmark and states for each agent. The action space is 50-dimensional, with 5 dimensions per agent that control their motions and interactions with the landmarks.

**iGibson:** Shown in Fig. 7 (d), iGibson has 42-dimensional observation space consisting of 4 state factors, including [Agent Location, Electric Appliances State, Object(s) in View, Robot Proprioception]. The action space is 11-dimensional, consisting of base velocity (2D), head motion (2D), arm motion (6D), and gripper motion (1D).

# D   Downstream Tasks

**DMC-Walker (Walker):**

- **Run**: In this task, the walker agent is rewarded for moving forward at a particular velocity.

- **Goal Reaching**: In this downstream task, the agent has to reach randomly generated goal positions.

**2D Gunner (2DG):**

- **Unlimited Ammo (unlim)**: In this downstream task, a set of targets will randomly appear, where the agent needs to navigate to a position close to the target and shoot them in order to score. The ammo is unlimited so the agent does not need to worry about picking up ammo.

- **Limited Ammo (lim)**: This downstream task is different from the "unlimited ammo" in that the agent starts with no ammo and needs to pick up ammo in order to shoot. Everything else is identical.

**Multi-Particle (MP):**

- **Sequential interaction (seq)** (easy, medium, hard): In this task, agents need to sequentially interact with their corresponding station following an instruction sequence given at the start of each episode. Interacting with stations in the wrong order will be penalized. The easy version of this task has a sequence length of 2, while medium and hard have a sequence length of 5 and 8 respectively.

- **Food-poison (fp)** (easy, medium, hard): In this downstream task, each station will offer either food or poison to the corresponding agent. Each agent needs to decide whether to interact with its corresponding station based on a sequence of binary indicators provided to the agents. The difficulty level has the same meaning as in the sequential interaction task.

**iGibson (IG):**

- **Look around**: In this task, the robot needs to look at objects in the room sequentially.

- **Appliances inspection**: In this task, the robot needs to navigate to different electric appliances, and test whether each of them is working correctly by pointing a remote control towards it.

- **Housekeeping**: In this task, the robot needs to manage the electric appliances intelligently. Specifically, the robot needs to first look at a screen to receive instructions. Depending on the instruction, the robot needs to turn on / off certain electric appliances using the remote control.

# E    Baseline Methods

During downstream task evaluation, we compared against the following state-of-the-art unsupervised RL methods:

- **DIAYN** [13] represents skill variable $z$ as an integer between 1 to $k^N$ and learns skills by maximizing $I(\mathcal{S}; \mathcal{Z})$, the MI between $\mathcal{Z}$ and all state factors $\mathcal{S}$.
- **CIC** [24] learns a state representation with contrastive learning and learns skills by maximizing transition entropy in the representation space.
- **CSD** [35] learns skills maximizing distance traveled along the direction of $z$ in the state space, where distance is measured in a controllability-aware manner.
- **METRA** [36] learn a set of behaviors that collectively cover as much of the state space as possible through optimizing a Wasserstein variant of the state-skill Mutual Information.
- **ICM** [37]: encourages visiting novel states by using prediction errors of action consequences as intrinsic rewards.
- **RND** [5] encourages visiting novel states by using prediction errors of features computed from a randomly initialized network as intrinsic rewards.
- **ELDEN** [17] operates in a factored state space similar to our approach, and encourages visiting states that induce novel factor dependencies.
- **SAC** [14] where no pretraining is used, and vanilla RL is directly applied to tackle the downstream tasks.

# F    Evaluating Skill Disentanglement Details

The DCI metric consists of three terms, namely **disentanglement**, **completeness**, and **informativeness**. In the context of this work, disentanglement (↑) measures, on average, to what extent each skill component only affects a single state factor. Completeness score (↑) measures, on average, to what extent each state factor is only influenced by a single skill component. Informativeness score (↑) measures the repeatability of learned skills: given the skill $z$, how accurately we can predict which states will be visited. We refer the reader to the work by Eastwood and Williams [12] for a detailed discussion of these metrics and how they are calculated.

# G    Hyperparameters

**Skill Dimensions:** For all skill learning methods with discrete skills (i.e. DUSDi, DIAYN), we make sure that they have equivalent capacity. Specifically, for igibson and 2D gunner, each DUSDi skill consists of 3 skill components, each component with 5 possible values. As a result, DIAYN skill is an integer between 1 to 125 in these two domains. The only exception is Multi-Particle, where DUSDi has ten sub-skills, each with 5 possible values. Since skill as an integer between 1 and $5^{10} = 9765625$ is obviously challenging for DIAYN to converge, we set the number of discrete skills to be 4096 for DIAYN. For continuous skills (i.e. CSD, CIC, METRA), we follow the skill dimensions specified in the original papers (64D for CIC, 3D for CSD and METRA), which were shown to be effective for the respective methods.

**Skill Learning Parameters:** All skill learning methods in our baselines use SAC to optimize for the intrinsic reward, with the same policy and value network architecture. DUSDi applies additional decomposition and masking to the value networks, as described in Section 3.2, which is not applicable to the baseline methods. Due to Q-decomposition, when using the same value network architecture, DUSDi's value network capacity is $N$ times of the capacity of other methods' value networks (including when comparing the variations of DUSDi, i.e., no decomposition). For a fair comparison, we also tried to increase value network capacity for other methods to match the capacity for DUSDi, but found that their skill/task learning performances do not improve significantly. This suggests (1) that, for skill learning, reward variance, rather than network capacity, is the key reason for no Q-composition variation of DUSDi to converge slowly, and (2) that, for task learning, disentangled skills, rather than network capacity, is what make DUSDi significantly outperform baselines.

We present the hyperparameters for SAC in Table. 2. All methods use a low-level step size of $L = 50$.

Table 2: Hyperparameters of Skill Learning.

|  | Name | Value |
|---|---|---|
|  | optimizer | Adam |
|  | activation functions | ReLu |
|  | learning rate | $1 \times 10^{-4}$ |
|  | batch size | 1024 |
| SAC | critic target $\tau$ | 0.01 |
|  | MLP size | [1024, 1024] |
|  | steps per update | 2 |
|  | # of environments | 4 |
|  | Temperature $\alpha$ | 0.02 |
|  | log std bounds | [-10, 2] |

**Downstream Hierarhical Learning:** For all skill discovery methods, downstream learning of the skill selection policy is implemented with PPO. We used the same hyperparameters for all methods across all tasks, as specified in Table. 3.

Table 3: Hyperparameters of Downstream Learning.

|  | Name | Value |
|---|---|---|
|  | optimizer | Adam |
|  | activation functions | Tanh |
|  | learning rate | $1 \times 10^{-4}$ |
|  | batch size | 32 |
|  | clip ratio | 0.1 |
| PPO | MLP size | [128, 128] |
|  | GAE $\lambda$ | 0.98 |
|  | target steps | 250 |
|  | n steps | 20 |
|  | # of environments | 4 |
|  | # of low-level steps $L$ | 50 |

**Downstream Finetuning:** For all non-skill discovery methods, downstream learning is done using the same hyperparameters as pretraining (table. 2), replacing the intrinsic reward with the task reward.

## H  Behavior Restriction of Skills via Domain Knowledge

Due to the decomposable nature of the intrinsic reward of DUSDi, we can conveniently restrict the behavior of skills by constraining the skill predictor $q_\phi^i(z^i|s^i)$ for a particular state factor $i$. For example, if we want $s^i$ to stay within a certain range, we can set $q_\phi^i(z^i|s^i)$ to be a uniform distribution for all $s^i$ not within this range, effectively discouraging the agent from going out of range. In the extreme case, we can fully specify the mapping between $z^i$ and $s^i$, essentially resulting in performing goal-conditioned RL for state $i$ (as also pointed out in [7]) while performing DUSDi for the rest of the state factors.

We qualitatively examine this idea in the iGibson domain. By restricting a mobile manipulator to only locomote in regions that are close to a whiteboard, our robot successfully learns diverse board-wiping behaviors which are otherwise extremely hard to learn. Visualizations of the learned skills can be seen on our project website.

