# OpenReview forum: "Disentangled Unsupervised Skill Discovery for Efficient Hierarchical Reinforcement Learning"
_NeurIPS.cc/2024/Conference — NeurIPS 2024 poster_

### Official Review · Reviewer_hssy · 2024-07-04

**Soundness:** 2
**Presentation:** 3
**Contribution:** 2
**Rating:** 5
**Confidence:** 3

**Summary:**

This paper proposes a method for learning disentangled skills that can be efficiently reused to solve downstream tasks. The mutual information objective design is simple yet effective. Intensive empirical results show the superiority of the proposed method.

**Strengths:**

The algorithm design, based on factored MDP, is well motivated. The empirical study shows that DUSDi consistently outperform other unsupervised skill discovery and unsupervised RL methods.

**Weaknesses:**

(a) The algorithmic contribution is not significant compared with other MI-based unsupervised skill discovery methods.

(b) The key objective design lacks theoretical support. As the author mentioned, Eq. (4) does not constitute a lower bound of the real objective.

(c) A qualitative study to show if the learned skill embedding is semantically detangled would be beneficial.

(d) The submitted code folder should include a readme file on how to reproduce the paper results.

**Questions:**

Please see the weakness part.

**Limitations:**

Provided.

---

> ### Author Rebuttal · Authors · 2024-08-06
>
> We thank the reviewer for the detailed reading of our paper and very constructive suggestions!
>
> >  The algorithmic contribution is not significant compared with other MI-based unsupervised skill discovery methods.
>
> We believe simple ideas that work well are very valuable for the community. As we demonstrate in our experiments, the algorithmic contribution of DUSDi, while may be considered simple, is highly effective and novel.
> Moreover, our contribution is orthogonal to most of the previous unsupervised skill discovery works and can be combined (as we discussed in the conclusion section).
>
> > The key objective design lacks theoretical support. As the author mentioned, Eq. (4) does not constitute a lower bound of the real objective.
>
> First, even though it is not a lower bound, our objective is still a plausible approximation of the true objective, and will be accurate when q is equal to p (line 142-143). More importantly, our empirical evaluations show that this objective works very well in practice; we explained and discussed in the paper the possible reasons (line 151-152). Lastly, we would like to point out that similar approximations have also been used in previous works [1].
>
> [1] Baumli, Kate, et al. "Relative variational intrinsic control." Proceedings of the AAAI conference on artificial intelligence. Vol. 35. No. 8. 2021.
>
> > A qualitative study to show if the learned skill embedding is semantically detangled would be beneficial.
>
> We agree with the reviewer that understanding (and visualizing) the disentangled nature of our learned skills is important to understanding the contribution of our work. However, we consider our manuscript already contains elements for that:
> We would like to respectfully point out that we provided visualization of the skills on our (anonymized) project website, with a special emphasis on the disentanglement of skills; we refer to the website multiple times in the paper (lines. 240 & footnote on page 1). We understand that reviewers are not forced to check these additional materials but due to space constraints, we are not able to include these visualizations in the main paper. We kindly refer the reviewer to our website for visualizations of the effect of each skill.
> In section 4.2 of our paper, we also provide quantitative evaluations of the level of disentanglement of our learned skills, as shown by Table. 1.
>
> > The submitted code folder should include a readme file on how to reproduce the paper results.
>
> We would like to respectfully point out that we have provided a README file with instructions on how to run our code, as in “/neurIPS-DUSDi/README.md” in the supplementary material. Please let us know if you have further suggestion about the content of the README file, and we will be working on polishing it for the next version of this paper.

---

> > ### Comment · Reviewer_hssy · 2024-08-12
> >
> > Thank you for your detailed feedback. I will maintain my rating, as only minor revisions have been made. The technical contribution and novelty of this paper are still borderline. Also, I suggest providing codes and instructions to reproduce ALL results in the paper.

---

### Official Review · Reviewer_JxTV · 2024-07-11

**Soundness:** 2
**Presentation:** 2
**Contribution:** 2
**Rating:** 5
**Confidence:** 3

**Summary:**

The paper presents a method (DUSDi) for learning reusable skills through unsupervised interactions. Unlike existing methods that produce entangled skills, DUSDi focuses on disentangling skills into components that each affect only one factor of the state space. This enables efficient chaining of skills via hierarchical reinforcement learning. A mutual-information-based objective ensures skill disentanglement, and value factorization optimizes the process.

**Strengths:**

1. The paper is well-written, and the empirical results show consistent advantages over baselines
2. The idea is simple and effective on the tested environments.

**Weaknesses:**

1. The paper assumes that the skill space is discrete; there is little information on how the formulations generalize to continuous skill space.
2. There is a lack of visualization of the learned skill embeddings, and how different skills affect specific factors of the state space, highlighting the disentangled nature, which makes it hard to determine the contribution of the proposed method to the overall performance gain over various baselines.

**Questions:**

Please address my concerns in the Weakness section

---

> ### Author Rebuttal · Authors · 2024-08-06
>
> We thank the reviewer for the detailed reading of our paper and very constructive suggestions!
>
> > there is little information on how the formulations generalize to continuous skill space
>
> For continuous skills, we can simply define each skill component z^i as a m-dimensional continuous vector (and therefore the length of the skill would be m*N, where N is the total number of skill components).
> Now we can define the prior distribution for each skill component distribution p(z^i) as an m-dimensional continuous uniform distribution (e.g. U[-1, 1]).
> Notice that our objective remains unchanged, as the MI is well-defined no matter whether the skill is discrete or continuous.
> Lastly, we need to change the output head of our skill prediction network, such that instead of outputting a categorical distribution, it will output a continuous probability distribution (e.g. a multivariate gaussian distribution with diagonal covariance).
> As a proof-of-concept, we have implemented a continuous version of our method with the settings described above, and examined it on 2DG domain, where the results are shown below:
>
> Task     | DUSDi-discrete | DUSDi-continuous |
> ------------|----------------|------------------|
> 2DG-unlim           | 39.7±0.4       | 39.7±0.6         |
> 2DG-lim         | 30.4±1.7       | 29.9±2.2         |
>
> As we can see, the continuous version of our method has similar performances as the discrete one on the 2DG domain, showcasing that we can indeed extend DUSDi to continuous skill space.
> We thank the reviewer for pointing this out, and we will add this content to the appendix for the next version of this paper.
>
> > There is a lack of visualization of the learned skill embeddings, and how different skills affect specific factors of the state space, highlighting the disentangled nature
>
> We agree with the reviewer that understanding (and visualizing) the disentangled nature of our learned skills is important to understanding the contribution of our work. However, we consider our manuscript already contains elements for that:
> we would like to respectfully point out that we provided visualization of the skills on our (anonymized) project website, with a special emphasis on the disentanglement of skills; we refer to the website multiple times in the paper (lines. 240 & footnote on page 1). We understand that reviewers are not forced to check these additional materials but due to space constraints, we are not able to include these visualizations in the main paper. We kindly refer the reviewer to our website for visualizations of the effect of each skill.
> In section 4.2 of our paper, we also provide quantitative evaluations of the level of disentanglement of our learned skills, as shown by Table. 1.
> With respect to the “visualizations of the skill embeddings” we would like to remark that in our work the skill variables themselves are discrete vectors. Beyond our visualizations of the (disentangled) effect of each skill, we are not sure what kind of visualization would show the “skill embeddings”, but we would gladly consider any suggestion the reviewer may have in mind.

---

> ### Comment · Reviewer_JxTV · 2024-08-14
>
> Thank you addressing my concerns. I will maintain my rating.

---

### Official Review · Reviewer_QWMX · 2024-07-11

**Soundness:** 4
**Presentation:** 4
**Contribution:** 3
**Rating:** 7
**Confidence:** 4

**Summary:**

Grounding on previous work on mutual information for skill learning, DUSDi proposed an algorithm for disentangled skill learning, which introduces several advances. First, DUSDi proposes to map factorized state components to factorized skill components. Then, it proposes the adoption of Q decomposition and Causal Policy Gradient to further enhance stability and efficiency.

**Strengths:**

The work presents a stable and efficient approach for unsupervised skill discovery, which presents the following strengths:
* **Simplicity**: the idea is straightforward and doesn't introduce complex components. Actually, through Q decomposition, DUSDi simplifies the learning of multiple skills.
* **Empirical study**: the empirical study is extensive, it includes modern baselines, such as CSD and METRA and it highlights well the strengths of the approach.
* **Presentation**: the presentation of the work is clear and of good quality. The additional visualizations on the website are interesting and further demonstrate the approach works as expected.

**Weaknesses:**

* **State space factorization**: the approach works well thanks to the assumption that the state space is well-factorized. However, for POMDPs or high-dimensional environments, the algorithm may not work as expected. The authors show that object-centric approaches can obviate such weakness. This works well for simple environments where colours can easily help distinguish objects, e.g. the multiparticle env, but it may not work as well in more complex environments.

**Questions:**

* the authors explicitly state that their objective is not optimizing a lower bound (line 150). Why not trying to actually keep a bound? e.g. following [1]
* since the authors discuss the unsupervised RL benchmark, and they use a similar setup, why not presenting results on the popular benchmark? I understand their point about the fact that these environments mostly require manipulating a single variable, but being a more standard benchmark, it would have been useful to see complete results on it (not just in the Walker env)
* what happens if you don't omit proprioceptive states from the MI optimization? (line 264-265)


[1] On Variational Bounds of Mutual Information, B. Poole et al,

**Limitations:**

The limitations of the approach are addressed in the paper.

---

> ### Author Rebuttal · Authors · 2024-08-06
>
> We thank the reviewer for the detailed reading of our paper and very constructive suggestions!
>
> > for POMDPs or high-dimensional environments, the algorithm may not work as expected.
>
> We agree with the reviewer that extending our method to partially observable complex pixel environments is non-trivial, and thus a great direction for future work that is opened up by this paper. For such complex domains, we may have to extend DUSDi with some form of explicit representation or memory, or with some implicit stateful architecture, combined with more powerful visual representation techniques such as SAM2 [1]. We will add this direction into the future work section in the next version of this paper.
>
> [1] Ravi, N., Gabeur, V., Hu, Y.-T., et al. (2024). SAM 2: Segment Anything in Images and Videos. arXiv preprint.
>
>
>
> > Why not trying to actually keep a bound?
>
> We thank the reviewer for pointing out this excellent paper. Since the objective we are trying to optimize contains negative signs in front of some MI terms, if we want to derive a lower bound for our objective, we need to derive upper bounds for the “negative MI terms”, As pointed out in [2], upper bounding MIs is more challenging than lower bounding it, and often involves more complicated expressions, which is why we ended up deciding to use lower bounds for all the MI terms. That being said, it is certainly interesting for future work to examine whether we can actually lower bound our objective in a computationally tractable way, and how that will affect the quality of learned skills. We will add this direction into the future work section in the next version of this paper.
>
> [2] On Variational Bounds of Mutual Information, B. Poole et al
>
>
>
> > I understand their point about the fact that (the standard) environments mostly require manipulating a single variable, but being a more standard benchmark, it would have been useful to see complete results on it (not just in the Walker env)
>
> We thank the reviewer for acknowledging our reasons for devoting our computing resources to more informative domains. Below, we provide some preliminary results for running on the quadruped domain  in URLB, where we follow exactly the setup of URLB, with 2×10^6 frames of pretraining and 1×10^5 frames of fine-tuning:
>
> Task| DDPG   | ICM         | RND         | DIAYN         | DUSDi  |
> ----------|--------|-------------|-------------|-------------|--------|
> Jump    | 236±48 | 205±33 | 590±33 |578±46 | 704±42 |
> Run    | 157±31 | 133±20 |  462±23 |415±28 | 450±13 |
> Stand  | 392±73 | 329±58 |  804±50 |706±48 | 844±26 |
> Walk | 229±57 | 143±31 |  826±19 |406±64 | 706±57 |
>
> Notice that baseline results are directly obtained from the URLB paper [3]. The table includes the returns obtained by each method. Despite not being the type of problem DUSDi is optimized for, we observe that DUSDi obtains best performance in 2 tasks, and performance comparable to the best solution in the others.
>
> [3] Laskin, Michael, et al. "Urlb: Unsupervised reinforcement learning benchmark." (2021).
>
> > what happens if you don't omit proprioceptive states from the MI optimization?
>
> The main reason that previous works like DIAYN and DADS omit proprioceptive states from the MI optimization is to prevent the agent from focusing too much on “less meaningful” states such as body postures, and we simply follow this setup since it is an easy heuristic to focus the learning on more meaningful explorations.
> Compared to methods like DIAYN, we anticipate our method to be less affected by adding the proprioceptive states into the MI optimization, since while one component of our objective will now be focusing on diversifying the less meaningful body posturing, the other components of our objective can still facilitate the mastering of more meaningful skills.

---

> ### Comment · Reviewer_QWMX · 2024-08-12
>
> I am happy with the author's rebuttal.
>
> I recommend they add some of the insights/comments provided here to the paper, as these may be useful for future readers, to have better insights into their work and its extendability.
>
> I am also glad they are adding experiments on the Quadruped domain of URLB, where I see their method obtains satisfactory performance (though not very outstanding).
>
> I will keep my score and recommend acceptance of the work.

---

> > ### Author Response · Authors · 2024-08-14
> >
> > We would like to express our gratitude for carefully reading our paper and providing valuable feedback. These suggestions are very constructive and have definitely helped us improve the quality of our paper in meaningful ways. Thank you!

---

### Official Review · Reviewer_bkGy · 2024-07-15

**Soundness:** 2
**Presentation:** 3
**Contribution:** 2
**Rating:** 4
**Confidence:** 3

**Summary:**

This paper proposed DUSDi, which learned disentangled skills in an unsupervised manner. Specifically, DUSDi learns each skill component only affects one factor of the state space. DUSDi is evaluated in a specialized tasks comparing with skill discovery methods.

**Strengths:**

- The proposed concept of disentanglement is novel
- the paper is easy to follow.

**Weaknesses:**

- It is not convincing how the disentanglement is beneficial in real-world scenarios.
- The empirical evaluation is limited; standard benchmarks (e.g., D4RL) is not used, and it is questionable whether it can be used in more complex environments like pixel-based environments

**Questions:**

See weaknesses.

**Limitations:**

See weaknesses.

---

> ### Author Rebuttal · Authors · 2024-08-06
>
> We thank the reviewer for the detailed reading of our paper and constructive suggestions.
>
> > It is not convincing how the disentanglement is beneficial in real-world scenarios.
>
> First, we would like to point out the realism of our experiments. We used iGibson, one of the benchmarks that best match the complexity of real-world robotics tasks. The action space in iGibson is exactly the same as in a real-world mobile manipulator with manipulation, navigation, and active sensing capabilities, and previous works have demonstrated that policies learned in iGibson can transfer zero-shot to the real world [1, 2]. The performance of DUSDi in iGibson and the other experiments is a strong support of the benefits of our method in real-world scenarios.
> Second, as discussed in the introduction, disentanglement is particularly helpful when dealing with complex state spaces consisting of many factors. This situation often appears in real-world applications such as complex robot systems (e.g., mobile manipulators), multi-object environments, or multi-agent systems, rendering DUSDi a critical solution for common scenarios.
>
> [1] Li, Chengshu, et al. "igibson 2.0: Object-centric simulation for robot learning of everyday household tasks." CoRL 2021.
>
> [2] Hu, Jiaheng, et al. "Causal policy gradient for whole-body mobile manipulation." RSS 2023.
>
> > standard benchmarks (e.g., D4RL) is not used
>
> We would like to point out that D4RL is a dataset for offline reinforcement learning, while our method, as well as all the baselines we compared against, are fully online. We do compare on standard online RL benchmarks as in DMC-walker (which is similar to the mujoco environment in D4RL), and discuss why we do not further test on more of those domains in section 4.1.
>
> > it is questionable whether it can be used in more complex environments like pixel-based environments
>
> We would like to respectfully point out that we have conducted experiments in pixel-based environments, and the entire section 4.5 is devoted to that set of experiments, where our results indicate that DUSDi can be used in pixel-based environments.

---

### Author Response · Authors · 2024-08-11

We would like to friendly remind the reviewers that our rebuttal contents and additional results can be seen in replies to each individual reviewer. We hope our rebuttal addresses your concern, and please kindly let us know if you have any further questions.

---

### Decision · Program_Chairs · 2024-09-25

**Decision:**

Accept (poster)

**Comment:**

The paper introduces DUSDi, a method for learning disentangled skills in unsupervised environments that can be efficiently reused and chained to solve downstream tasks. The idea presented in the paper is significant, well-executed, and introduces novelties around the disentanglement of skills which could potentially be scaled to more complex settings. The primary concern is that the key novelty (Equation 4) should be more strongly supported from a theoretical standpoint, although the authors provide a reasonable justification surrounding this point, including previous works which have incorporated similar approximations. Apart from this, the discrete nature of skill spaces is fairly common, and is as such, not viewed as a major limitation/flaw, although means to extend the idea to continuous skill spaces is welcome. The proposed method also demonstrates that it can be applied in pixel-based environments, which demonstrates its general applicability. Although D4RL was suggested, it is a baseline commonly employed for offline methods and is as such, not applicable to the method in question.